# OmniX: From Unified Panoramic Generation and Perception to Graphics-Ready 3D Scenes

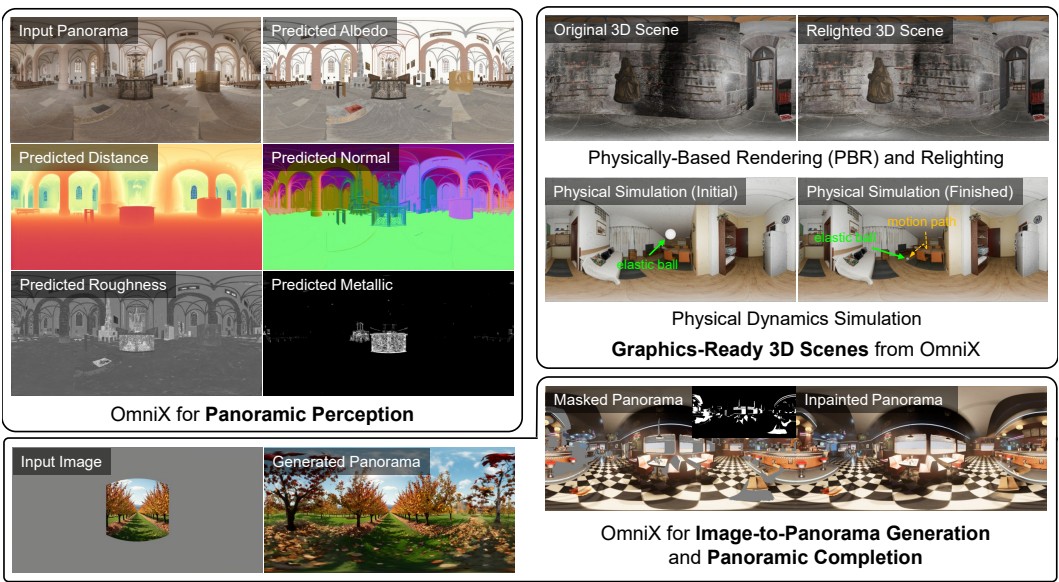

Figure 1: We present **OmniX**, a versatile and unified framework that repurposes pre-trained 2D flow matching models for panoramic perception, generation, and completion. This framework enables the construction of immersive, photorealistic, and graphics-compatible 3D scenes, suitable for physically-based rendering (PBR), relighting, and physical dynamics simulation.

## Abstract

There are two prevalent ways to constructing 3D scenes: procedural generation and 2D lifting. Among them, panorama-based 2D lifting has emerged as a promising technique, leveraging powerful 2D generative priors to produce immersive, realistic, and diverse 3D environments. In this work, we advance this technique to generate graphics-ready 3D scenes suitable for physically based rendering (PBR), relighting, and simulation. Our key insight is to repurpose 2D generative models for panoramic perception of geometry, textures, and PBR materials. Unlike existing 2D lifting approaches that emphasize appearance generation and ignore the perception of intrinsic properties, we present OmniX, a versatile and unified framework. Based on a lightweight and efficient cross-modal adapter structure, OmniX reuses 2D generative priors for a broad range of panoramic vision tasks, including panoramic perception, generation, and completion. Furthermore, we construct a large-scale synthetic panorama dataset containing high-quality multi-modal panoramas from diverse indoor and outdoor scenes. Extensive experiments demonstrate the effectiveness of our model in panoramic visual perception and graphics-ready 3D scene generation, opening new possibilities for immersive and physically realistic virtual world generation.

## 1 Introduction

Digitizing the 3D world we live in is a technological endeavor that is both imaginative and valuable. Digital replication Dai et al. (2024); Huang et al. (2025b) of 3D scene allows us humans to obtain

entertainment and interactive experiences that are difficult to obtain in daily life, or enables near-zero-cost simulation learning for intelligent agents or robots. However, constructing complex 3D scenes requires significant effort and time from artists and engineers, which limits the scale of 3D scene data and hinders the development of native 3D scene generative models.

To automatically build 3D scenes while circumventing data shortages, the community has leveraged large visual language foundation models trained on large-scale text, image, and video data. Based on these powerful models, two typical approaches emerge: procedural generation (Raistrick et al., 2023b;a; Feng et al., 2023) and 2D lifting (Lee et al., 2024; Yu et al., 2024b;c). While procedural generation relies on retrieving objects from a 3D asset library to build the scene, 2D lifting methods directly repurpose 2D generative priors for 3D scene generation, achieving diverse and high-quality results. Recent works (Yang et al., 2025; Huang et al., 2025a; Li et al., 2024b; Zhou et al., 2024) further introduces panoramic representations, which serve as a bridge between 2D and 3D, greatly improving the cross-view consistency of generated 3D scenes. However, these works emphasize appearance generation rather than intrinsic perception, generally using off-the-shelf depth estimation models to extract scene geometry without textures and PBR materials. This hinders the integration of generated 3D scenes into modern graphics pipelines.

In this paper, we introduce **OmniX**, a versatile framework repurposing pre-trained 2D flow matching models for panoramic generation, intrinsic perception, and masked completion. First, we establish a unified formulation for different vision tasks, redirecting the 2D generative paradigm for image-to-panorama generation, panorama-to-X perception, and their generalization with mask guidance. Furthermore, we explore different cross-modal adapter structures capable of handling multiple inputs and propose an effective and flexible adapter structure. This structure can fully reuse 2D generative priors for different vision tasks without significantly changing the pre-trained model weights, effectively improving the performance of panoramic visual perception. In addition, we construct a synthetic panoramic dataset, PanoX, covering indoor and outdoor scenes and various visual modalities such as distance, normal, albedo, roughness, and metallic. This dataset addresses the shortage of high-quality panoramic data with dense geometry and material annotations.

Our main contributions are as follows:

- We present OmniX, a versatile framework repurposing pre-trained 2D flow matching models for panoramic generation, perception, and completion. With a unified formulation and effective cross-modality adapter design, we demonstrate the potential of OmniX to unify panoramic generation, perception, and completion.

- We introduce PanoX, a synthetic panorama dataset encompassing both indoor and outdoor scenes, along with various visual modalities such as depth, normal, albedo, roughness, and metallic maps. This dataset addresses the gap in high-quality panoramic data featuring dense geometry and material annotations.

- Extensive experiments demonstrate the effectiveness of our approach in panoramic perception, generation, and completion. Our method further enables the construction of immersive, photorealistic, and graphics-compatible 3D scenes, ready for PBR rendering, relighting, and physical simulation.

## 2 RELATED WORK

### 2.1 INVERSE RENDERING

Inverse rendering (Barrow et al., 1978; Barron & Malik, 2014; Bousseau et al., 2009; Bell et al., 2014; Bhattad et al., 2023; Grosse et al., 2009; Li & Snavely, 2018; Li et al., 2020; Liang et al., 2023; Sengupta et al., 2019; Wang et al., 2021; Wimbauer et al., 2022) aims to estimate intrinsic scene properties such as geometry, materials, and lighting from images. With the rapid progress of generative models, particularly diffusion models, researchers have explored their potential for inverse rendering (Kocsis et al., 2025; Li et al., 2025; Liang et al., 2025; Zeng et al., 2024; Kocsis et al., 2024; Zhu et al., 2022; Li et al., 2022; 2018). IntrinsiX (Kocsis et al., 2025) generates high-quality PBR maps (albedo, roughness, metallic, normal) from text prompts using a diffusion process, supporting precise material and lighting editing. DiffusionRenderer (Liang et al., 2025) leverages

video diffusion models for joint inverse and forward rendering, combining G-buffer estimation with photorealistic image generation through co-training on synthetic and real data.

Panoramic images capture a wider field of view and provide more comprehensive scene information, making them versatile for various applications. Yet, inverse rendering with panoramas remains underexplored. PhyIR (Li et al., 2022) recovers geometry, complex SVBRDFs, and spatially-coherent illumination from a panoramic indoor image using an enhanced SVBRDF model and a physics-based in-network rendering layer to handle complex materials like glossy, metal, and mirror surfaces. However, it is limited to indoor scenes, while we leverage 2D generative priors to generalize across both indoor and outdoor environments.

### 2.2 3D Scene Generation

Procedural generation (Parish & Müller, 2001; Musgrave et al., 1989; Cordonnier et al., 2017; Raistrick et al., 2023b;a; Yu et al., 2011; Deitke et al., 2022; Feng et al., 2023) automatically creates 3D scenes based on predefined rules or constraints. These methods are scalable and widely used in domains such as gaming, urban planning, and architecture, but often lack diversity and realism due to their rule-based nature. Representative works include CityEngine (Parish & Müller, 2001), which uses grammar-based rules for urban layouts, and Infinigen (Raistrick et al., 2023a), which integrates terrain, material, and creature generators to produce diverse natural environments.

Image- and video-based methods bridge 2D inputs and 3D representations (Dastjerdi et al., 2022; Tang et al., 2023; Lee et al., 2024; Yu et al., 2024a; Li et al., 2024a; Zhang et al., 2024a; Li et al., 2023b; Höllein et al., 2023; Zhang et al., 2024b; Yu et al., 2024d). Image-based approaches reconstruct 3D scenes from single or sequential images using outpainting or depth estimation, with works like ImmerseGAN (Dastjerdi et al., 2022) and MVDiffusion (Tang et al., 2023) generating panoramas for scene synthesis. Video-based methods leverage temporal information to ensure coherent dynamic scenes, exemplified by VividDream (Lee et al., 2024) and 4Real (Yu et al., 2024a). These methods emphasize appearance generation, relying on off-the-shelf depth estimators for geometry while neglecting intrinsic properties such as albedos, normals, and PBR materials.

## 3 Method

### 3.1 Overview

We introduce OmniX, a versatile framework repurposing pre-trained 2D flow matching models (Esser et al., 2024; Lipman et al., 2023) for panorama perception and generation (Sec. 3.3), which facilitates graphics-ready 3D scene generation (Sec. 3.4). Additionally, we construct a multi-modal synthetic panorama dataset, PanoX, which will be introduced in Sec. 3.2.

### 3.2 PanoX: A Multimodal Synthetic Panorama Dataset

Omnidirectional visual perception is crucial for visual understanding and spatial intelligence. To effectively learn perception across a wide field of view (FoV), large-scale panoramic datasets with dense annotations are necessary. While several narrow FoV image datasets (Roberts et al., 2021; Zhu et al., 2022; Li et al., 2023c) offer rich geometry and material annotations, there remains a scarcity of panoramic datasets equipped with dense annotations within the research community.

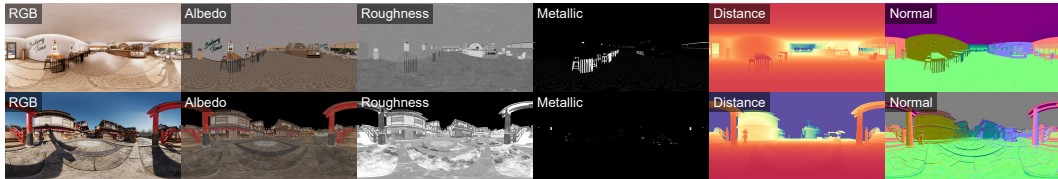

Figure 2: A preview of the proposed PanoX dataset, covering both indoor and outdoor scenes.

To this end, we introduce PanoX, a multimodal synthetic panorama Dataset with dense geometry and material annotations. Given the challenges of collecting real panoramic data and the high costs

of manual annotation, we leverage synthetic 3D scene assets and the Unreal Engine 5 to generate pixel-aligned multimodal panoramic data. Specifically, our dataset covers 8 large-scale 3D scenes (5 indoor and 3 outdoor scenes) like stores, warehouses, and wilderness. These scenes are rendered into RGB panoramas along with their distance maps, surface normals, albedo, roughness, and metallicity. A preview of PanoX is shown in Figure 2. We also provide text descriptions corresponding to the panoramic images, which are extracted using Florence 2 (Xiao et al., 2024). The entire dataset contains more than 10,000 instances, corresponding to 60,000 panoramic images of different modalities. We split the six PanoX scenes into training, validation, and test segments in an 8:1:1 ratio, resulting in PanoX-Train, PanoX-Val, and PanoX-Test. The remaining two scenes are grouped as PanoX-OutDomain for generalization evaluation.

We compare the proposed PanoX dataset with existing synthetic scene datasets, as shown in Table 1. To the best of our knowledge, the proposed PanoX is the first panoramic dataset covering both indoor and outdoor scenes with dense geometry and material annotations.

Table 1: Comparison between existing synthetic scene datasets and our proposed PanoX.

| Datasets | Geometry Annotation | Material Annotation | Panorama | Indoor | Outdoor |
|---|---|---|---|---|---|
| InteriorNet (Li et al., 2018) | ✓ | ✓ | ✓[†] | ✓ | ✗ |
| Structured3D (Zheng et al., 2020) | ✓ | ✗ | ✓ | ✓ | ✗ |
| Hypersim (Roberts et al., 2021) | ✓ | ✗ | ✗ | ✓ | ✗ |
| InteriorVerse (Zhu et al., 2022) | ✓ | ✓ | ✗ | ✓ | ✗ |
| FutureHouse[‡] (Li et al., 2022) | ✓ | ✓ | ✓ | ✓ | ✗ |
| MatrixCity (Li et al., 2023c) | ✓ | ✓ | ✗ | ✗ | ✓ |
| PanoX (Ours) | ✓ | ✓ | ✓ | ✓ | ✓ |

[†] InteriorNet only contains panoramic RGB images without geometry and material annotations.
[‡] FutureHouse is no longer publicly available.

### 3.3 OMNIX: UNIFIED PANORAMIC GENERATION, PERCEPTION, AND COMPLETION

OmniX is a versatile framework for unified panoramic perception and generation, built on the pretrained 2D flow matching model, FLUX.1-dev (Labs, 2025). Before delving into the technical details, we first provide a general formulation for unified visual perception and generation.

**Unified formulation.** Typically, a flow matching-based image generator $f_\theta$ is trained to predict the velocity vector $\mathbf{v}$ from latent representation $\mathbf{z}_0$ to latent representation $\mathbf{z}_1$, given a textual prompt $y$ and the current timestep $t$:

$$\mathbf{v}_t = f_\theta(\mathbf{z}_t, y, t), \tag{1}$$

The predicted target $\hat{z}_1$ can be obtained by solving the following ordinary differential equation:

$$\hat{\mathbf{z}}_1 = \mathbf{z}_0 + \int_0^1 \mathbf{v}_t \, dt = \mathbf{z}_0 + \int_0^1 f_\theta(\mathbf{z}_t, y, t) dt. \tag{2}$$

Our goal is to expand this image generation paradigm into a unified panoramic generation, perception, and completion framework and serve the subsequent 3D scene construction. To this end, we generalize the model $f_\theta$ to take multiple condition inputs:

$$\hat{\mathbf{z}}_1 = \mathbf{z}_0 + \int_0^1 f_\theta(\mathbf{z}_t, \mathbf{c}^0, \mathbf{c}^1, ..., y, t) dt, \tag{3}$$

where $\{\mathbf{c}^i | i = 0, 1, ...\}$ are the input conditions spatially aligned with $\mathbf{z}_t$. The modality and number of $\mathbf{c}^i$ depends on the specific task. We explore three task settings throughout this paper:

(**i**) For panoramic perception tasks, *i.e.*, RGB→X, we define $\mathbf{c}^0$ as the RGB reference, and set $y = \varnothing$. The target $\mathbf{z}_1$ can be any visual modality, such as depth (Euclidean distance for panoramas), normal, albedo, roughness, and metallic. Optionally, additional conditions can be provided to improve performance, for example, further defining $\mathbf{c}^1$ as a camera ray.

(ii) For panoramic completion tasks, we define $\mathbf{c}^0$ as the masked image, and $\mathbf{c}^1$ as the corresponding mask. Note that visual completion includes image-to-panorama generation, where the masked image is defined as the empty panorama with conditional view projected onto it.

(iii) For panoramic visual perception tasks, we define $\mathbf{c}^0$ as the RGB reference, $\mathbf{c}^1$ as the masked target, and $\mathbf{c}^2$ as the corresponding mask. The textual prompt $y$ is set to $\varnothing$. This task setting is necessary for progressive completion when building 3D scenes.

**Cross-modal adapter structure.** The flexibility of Diffusion Transformer (DiT) (Peebles & Xie, 2023) enables multiple ways to adapt the DiT-based flow matching model for multiple cross-modal 2D inputs, as shown in Figure 3. Specifically, depending on how branches and adapters are shared, these methods can be divided into: Shared-Branch, Shared-Adapter, and Separate-Adapter. In subsequent experiments, we demonstrate that the Separate-Adapter architecture achieves best visual perception performance (Table 4) and allows flexible expansion of inputs and outputs without significantly changing the distribution of model weights.

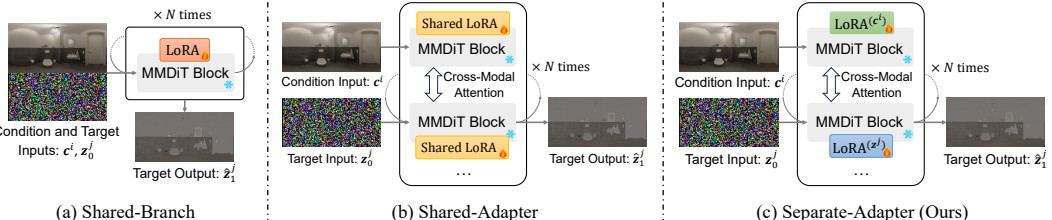

(a) Shared-Branch                (b) Shared-Adapter                (c) Separate-Adapter (Ours)

Figure 3: Different cross-modal adapter structures, which enables multiple condition inputs $\{\mathbf{c}^i \mid i = 0, 1, ...\}$ and target outputs $\{\hat{\mathbf{z}}_1^j \mid j = 0, 1, ...\}$ from various visual modalities.

**Optimization.** Based on the Separate-Adapter architecture, multiple LoRAs (Hu et al., 2022) are optimized to leverage a pre-trained 2D flow matching model for feature extraction of condition inputs and velocity vector prediction of target outputs. While both the condition $\mathbf{c}$ and the target $\mathbf{z}_t$ are input to the DiT model, only the prediction of target are used to compute the flow matching loss Lipman et al. (2023):

$$\mathcal{L} = \mathbb{E}_{t, \mathbf{z}_1, \mathbf{z}_0} \|\mathbf{v} - f_\theta(\mathbf{z}_t, \mathbf{c}, t)\|^2, \tag{4}$$

where the velocity vector $\mathbf{v} = \mathbf{z}_1 - \mathbf{z}_0$. Note that this objective can be generalized to multiple conditional inputs $\{\mathbf{c}^i \mid i = 0, 1, ...\}$ and multiple target outputs $\{\mathbf{z}_1^j \mid j = 0, 1, ...\}$, yielding:

$$\mathcal{L}_{\text{mimo}} = \mathbb{E}_{t, \mathbf{z}_1^j, \mathbf{z}_0} \|\mathbf{v} - f_\theta(\mathbf{z}_t, \mathbf{c}^0, \mathbf{c}^1, ..., t)\|^2. \tag{5}$$

**Remarks.** Although this paper focuses on panoramic data, the OmniX framework established above can also be applied to narrow FoV images. We try not to introduce inductive bias of omnidirectional representations, thereby preserving the 2D generative priors of the pre-trained model. However, we empirically found that the DiT model has difficulty learning the seam continuity of ERP panoramas, which may be attributed to the topological limitations of the 2D position encoding. To this end, we follow LayerPano3D (Yang et al., 2025) and introduce the horizontal blending technique.

## 3.4 GRAPHICS-READY 3D SCENE GENERATION

Leveraging the OmniX framework, we are able to construct graphics-ready 3D scenes from a single image input. The entire pipeline consists of three stages: (a) Multimodal panorama generation, (b) Scene reconstruction, and (c) Interactive completion.

**Multimodal panorama generation.** The proposed OmniX framework offers a general solution for image-to-panorama generation and RGB-to-X panorama perception. We train multiple adapters to repurpose the pre-trained flow matching model for these tasks. Ultimately, by combining adapters of different tasks, we can achieve a generative chain of "Image → Panorama → Panorama with Intrinsic Properties".

**Scene reconstruction.** Given a panoramic distance map, since the ray direction corresponding to each pixel is known, the pixels can be projected into 3D space as vertices of a 3D mesh, following

DreamCube (Huang et al., 2025a). The connectivity of these vertices can be further determined based on pixel neighbors and relative distances. Once the 3D mesh of the scene is obtained, the panoramic maps of other modalities (*i.e.*, albedo, normal, roughness, and metallic) can be assigned to each triangle face via spherical UV unwrapping, resulting in a PBR-ready scene-level 3D asset.

**Interactive completion.** A single panoramic image is only an omnidirectional observation from a fixed position, so the reconstructed scene does not support free exploration. Interactive scene completion (Yang et al., 2025; Yu et al., 2024b) is important for constructing explorable and even city-scale 3D scenes. To this end, we enhance the OmniX adapters with mask input and fine-tune them for completion and guided perception, resulting in OmniX-Fill. In particular, to simulate scene holes caused by occlusion, we design a depth-based sampling technique to produce occlusion-aware masks, as illustrated in Figure 4. When interactively completing a scene, panoramic completion and guided perception allow generating new regions while preserving existing ones.

Figure 4: **Occlusion-aware mask sampling.** Based on the panoramic distance map and a randomly sampled 3D displacement, we can estimate the occluded regions by ray intersection. These regions are used as masks for training panoramic completion and guided panoramic perception models.

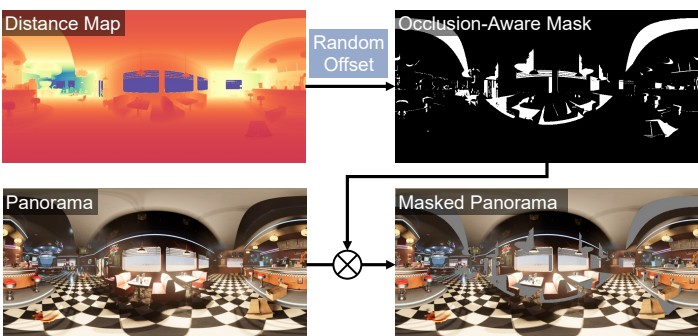

## 4 EXPERIMENT

### 4.1 IMPLEMENTATION DETAIL

We trained 12 adapter models based on Flux.1-dev for different vision tasks, including: OmniX-Image2Pano, OmniX-Pano2Depth, OmniX-Pano2Normal, OmniX-Pano2Albedo, OmniX-Pano2Roughness, OmniX-Pano2Metallic, and their corresponding OmniX-Fill versions. Each of the adapter model consists of two or more LoRAs, depending on the number of inputs and outputs.

Our method is implemented in PyTorch, trained and evaluated on four Ascend 910B NPUs with batch size of 1. All models use the same optimization settings. We adopt an AdamW optimizer with learning rate of 1e-4 for training. No learning rate decay strategy is employed. For graphics-related applications, we develop based on Blender 4.2 and deploy them on an Nvidia L40S GPU.

### 4.2 EXPERIMENTAL SETUP

**Datasets.** We utilize the proposed PanoX (Sec. 3.2), Structured3D (Zheng et al., 2020), and HDR360-UHD (Chen et al., 2022) datasets for training and evaluation. The details of PanoX are given in Sec. 3.2. Structured3D (Zheng et al., 2020) is a large-scale photo-realistic dataset, containing about 20,000 indoor panoramas with albedo, depth, normal, and semantic annotations. HDR360-UHD (Chen et al., 2022) is an HDR panorama dataset collected from online resources, covering both indoor and outdoor scenes. We convert these HDR images into LDR images and remove samples with invalid areas, resulting in thousands of high-quality RGB panoramas.

For panoramic perception, we use the standard training splits of both PanoX and Structured3D as training sets. Each training batch is sampled from these two data sources with equal probability. In particular, Structured3D does not include PBR materials, so only PanoX is used when training the roughness and metallic perception models. For panoramic generation, we use the entire HDR360-UHD dataset as the training set, where the text prompts are extracted by BLIP 2 Li et al. (2023a). All panoramic images are resized to a resolution of $512 \times 1024$ during training.

**Evaluation metrics.** For visual perception tasks with ground truths, we adopt a variety of quantitative metrics for different modalities. Specifically, we use PSNR (Peak Signal-to-Noise Ratio) and LPIPS (Zhang et al., 2018) as metrics for albedo, roughness, and metallic. For Euclidean distances, we use four commonly used metrics: AbsRel, $\delta$-1.25, MAE, and RMSE, following the implementation in Cheng et al. (2018). For surface normals, We measure the pixel-wise angular error with ground truth and report the mean, median, and the percentage of pixels with an error below 5° and 30° following Bae & Davison (2024). Note that there may be invalid values in the ground truths (*e.g.*, pixels at infinite distance), we exclude these invalid values when calculating the metrics.

### 4.3 RESULTS ON PANORAMIC PERCEPTION

We divide panoramic perception into intrinsic image decomposition (albedo, roughness, metallic) and geometry estimation (distance, normal), and present both qualitative and quantitative results compared to the state-of-the-art methods.

**Panoramic intrinsic decomposition.** We compare our OmniX with five state-of-the-art intrinsic decomposition methods: RGB↔X (Zeng et al., 2024), MGNet Zhu et al. (2022), IDArb Li et al. (2025), IID (Kocsis et al., 2024), DiffusionRenderer (Liang et al., 2025). Note that DiffusionRenderer is a video-based inverse rendering method, so we render each panorama into multiple frames to fit its input. The quantitative results are reported in Table 2. Our method achieves consistent state-of-the-art performance on the prediction of three intrinsic properties: albedo, roughness, and metallic. A qualitative comparison is shown in Figure 5 to illustrate the prediction results.

Table 2: **Quantitative evaluation of OmniX on panoramic intrinsic decomposition** compared to five competitors: RGB↔X (Zeng et al., 2024), MGNet (Zhu et al., 2022), IDArb (Li et al., 2025), IID (Kocsis et al., 2024), and DiffusionRenderer (Liang et al., 2025). For fair comparison, we use PanoX-OutDomain as the test set to ensure all methods are evaluated in unseen scenarios.

| Methods | Albedo | | Roughness | | Metallic | |
|---|---|---|---|---|---|---|
| | PSNR↑ | LPIPS↓ | PSNR↑ | LPIPS↓ | PSNR↑ | LPIPS↓ |
| RGB↔X | 6.347 | 0.591 | 8.175 | 0.628 | 4.384 | 0.720 |
| MGNet | 7.934 | 0.583 | 10.219 | 0.625 | 6.368 | 0.656 |
| IDArb | 9.420 | 0.562 | 9.572 | 0.603 | 4.296 | 0.554 |
| IID | 10.250 | 0.640 | 10.092 | 0.631 | 7.891 | 0.726 |
| DiffusionRenderer | 10.906 | 0.556 | 10.445 | 0.591 | 14.453 | 0.425 |
| OmniX | **17.755** | **0.344** | **16.211** | **0.398** | **18.874** | **0.254** |

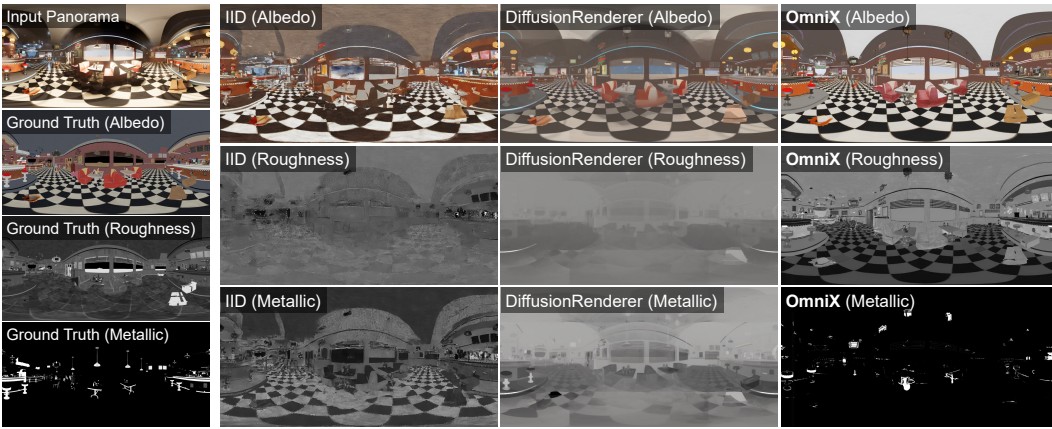

Figure 5: Qualitative evaluation of OmniX on panoramic intrinsic decomposition compared to state-of-the-art methods: IID (Kocsis et al., 2024), and DiffusionRenderer (Liang et al., 2025).

**Panoramic geometry estimation.** We compare our OmniX with two panoramic geometry esti- mation methods: DepthAnyCamera (Guo et al., 2025), DepthAnywhere Wang & Liu (2024), and four narrow-FoV geometry estimation methods: OmniData-v2 (Kar et al., 2022), MGNet Zhu et al. (2022), DiffusionRenderer (Liang et al., 2025), and MoGe (Wang et al., 2025). The quantitative results are reported in Table 3, where we achieve the highest normal estimation accuracy and and the second highest depth estimation accuracy. Note that MoGe (Wang et al., 2025) integrate 21 large-scale datasets for training, while we use much less data to achieve competitive performance. We further provide a qualitative comparison in Figure 6 to illustrate the prediction results.

Table 3: **Quantitative evaluation of OmniX on panoramic geometry estimation** compared to state-of-the-art methods: DiffusionRenderer (Liang et al., 2025), MGNet (Zhu et al., 2022), DepthAnywhere (Wang & Liu, 2024), OmniData-v2 (Kar et al., 2022), DepthAnyCamera (Guo et al., 2025), and MoGe (Wang et al., 2025). For fair comparison, we use PanoX-OutDomain as the evaluation set to ensure all methods are evaluated in unseen scenarios.

| Methods | Distance | | | | Normal | | | |
|---|---|---|---|---|---|---|---|---|
| | AbsRel↓ | δ-1.25↑ | MAE↓ | RMSE↓ | Mean↓ | Median↓ | 5°↑ | 30°↑ |
| DiffusionRenderer | 0.709 | 0.246 | 2.553 | 16.095 | 97.186 | 89.621 | 0.001 | 0.023 |
| MGNet | 0.433 | 0.396 | 3.972 | 11.321 | 79.955 | 82.836 | 0.019 | 0.269 |
| DepthAnywhere | 0.345 | 0.392 | 1.804 | 9.590 | / | / | / | / |
| OmniData-v2 | 0.342 | 0.440 | 1.944 | 10.763 | 85.220 | 100.596 | 0.150 | 0.245 |
| DepthAnyCamera | 0.199 | 0.680 | 1.930 | 7.858 | / | / | / | / |
| MoGe | **0.106** | **0.898** | **1.039** | **5.352** | / | / | / | / |
| OmniX | 0.158 | 0.787 | 1.680 | 6.828 | **27.138** | **14.879** | **0.155** | **0.663** |

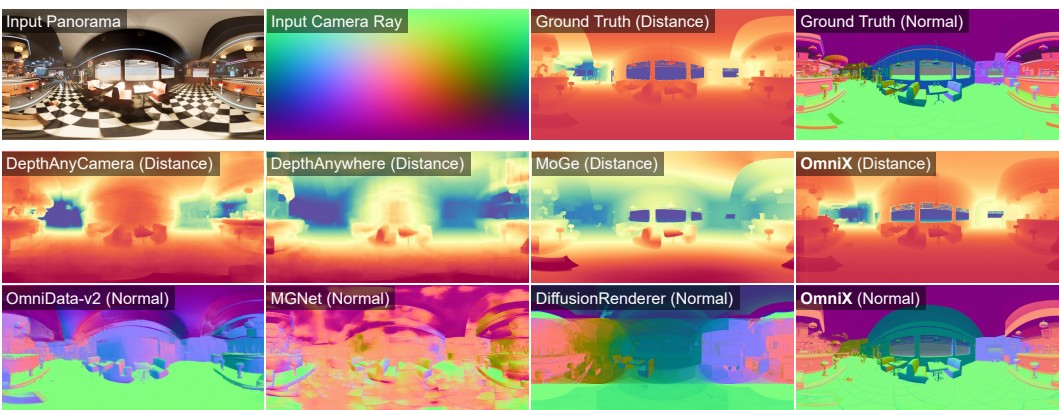

Figure 6: Qualitative evaluation of OmniX on panoramic geometry estimation compared to state-of- the-art geometry estimation methods: DepthAnyCamera (Guo et al., 2025), DepthAnywhere (Wang & Liu, 2024), OmniData-v2 (Kar et al., 2022), MGNet (Zhu et al., 2022), DiffusionRenderer (Liang et al., 2025), and MoGe (Wang et al., 2025). Our approach demonstrates a notable advantage in capturing fine image details, thanks to the proposed cross-modal adapter structure that effectively leverages 2D generative priors to enhance visual perception.

**In-the-wild panoramic perception.** We empirically find that the proposed OmniX demonstrates superior generalization performance and is able to achieve satisfactory prediction results on in-the- wild images from the Internet. These results are presented in the supplementary material.

## 4.4 ABLATION ANALYSIS AND DISCUSSION

We explore the impacts of adapter structures and camera rays on panoramic perception performance. Further ablation analysis and discussion are provided in the supplementary material.

**Impact of adapter structures.** We experimentally analyze the impact of different adapter architectures on panoramic perception performance, as shown in Table 4. The Separate-Adapter structure adopted by OmniX achieved the best performance, thanks to its full reuse of the 2D generative prior for feature extraction of the reference image and prediction of the target modality, without significantly changing the distribution of the pre-trained model weights.

Table 4: **Impact of adapter structures.** We utilize PanoX-Test and PanoX-OutDomain together as the evaluation set to comprehensively cover both in-domain and out-domain scenarios.

| Settings | Albedo | | Roughness | | Distance | | | |
|---|---|---|---|---|---|---|---|---|
| | PSNR↑ | LPIPS↓ | PSNR↑ | LPIPS↓ | $\delta$-1.25↑ | AbsRel↓ | RMSE↓ | MAE↓ |
| Shared-Branch | 15.294 | 0.650 | 11.729 | 0.667 | 0.464 | 0.386 | 8.565 | 2.122 |
| Shared-Adapter | 20.462 | 0.305 | 16.920 | 0.363 | 0.689 | 0.219 | 6.346 | 1.363 |
| Separate-Adapter (Ours) | **21.682** | **0.260** | **18.162** | **0.329** | **0.808** | **0.154** | **4.755** | **1.110** |

**Impact of camera ray inputs.** Camera rays are considered important for spatial perception and understanding. We investigate the impact of incorporating camera rays as additional input on visual perception performance. As reported in Table 5, the introduction of camera rays slightly improves the estimation accuracy of normal maps, but has no significant improvement on other modalities.

Table 5: **Impact of camera ray inputs.** We utilize PanoX-Test and PanoX-OutDomain together as the evaluation set to comprehensively cover both in-domain and out-domain scenarios.

| Settings | Distance | | Normal | | Albedo | | Roughness | Metallic |
|---|---|---|---|---|---|---|---|---|
| | AbsRel↓ | $\delta$-1.25↑ | Mean↓ | Median↓ | PSNR↑ | LPIPS↓ | PSNR↑ | PSNR↑ |
| w/o CamRay | **0.154** | **0.808** | 20.578 | 11.715 | **21.682** | **0.260** | **18.162** | 24.643 |
| w/ CamRay | 0.155 | **0.808** | **19.917** | **10.992** | 21.287 | **0.260** | 17.590 | **25.523** |

## 4.5 APPLICATIONS

OmniX enables automatic production of graphics-ready 3D scenes, as described in Section 3.4. To evaluate the practicality of these generated 3D scenes, we import them into Blender and implement various graphics workflows, including PBR-based relighting and physical simulation. Specifically, for **PBR-based relighting**, we add a point light source and animate its horizontal movement in a circular path around the scene's center. For **physical simulation**, we introduce an elastic ball into the scene, assigning it an initial horizontal velocity to enable dynamic interactions within the environment. Demonstration videos for these workflows are included in the supplementary material.

## 5 CONCLUSION

In this work, we present OmniX, a versatile framework for repurposing pre-trained 2D flow matching models for panoramic perception, generation, and completion. Specifically, we establish a unified formulation that incorporates dense visual perception (RGB→X) and visual completion (masked X→X) into a 2D generative paradigm, and propose an efficient and lightweight cross-modal adapter architecture to model diverse task-specific knowledge. Furthermore, we collect a synthetic panoramic dataset, PanoX, which covers indoor and outdoor scenes and various visual modalities. PanoX provides a panoramic perception benchmark for the community, addressing the shortage of panoramic data with dense geometry and material annotations. Comprehensive experiments validate the effectiveness of our approach in panoramic perception, generation, and completion. Additionally, our method facilitates the creation of immersive, photorealistic, and graphics-compatible 3D scenes, seamlessly integrating with PBR rendering, relighting, and physical simulation workflows.

ETHICS STATEMENT

This research involves the generation and perception of panoramas using publicly available or synthetic datasets. The proposed PanoX dataset relies exclusively on artificially created datasets, and there are no privacy concerns or risks associated with real-world data collection or human subjects. We have ensured that all data used in this study do not contain any personally identifiable information, and no ethical approval was required. We are committed to conducting our research in accordance with ethical standards and responsible AI development practices.

REPRODUCIBILITY STATEMENT

We have provided detailed descriptions of our implementation, including model architectures (Section 3.3), training procedures (Section 4.1), and data processing methods ((Section 4.2)), to facilitate reproducibility of our results. Additionally, we commit to open-sourcing our code, trained model weights, and datasets upon publication.

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

## A    IN-THE-WILD PANORAMIC PERCEPTION

We provide the panoramic perception results of OmniX on in-the-wild images from the Internet, as shown in Figure 7. Our method demonstrates excellent generalization performance for unseen images. This is due to our proposed adapter architecture effectively reusing the generative priors from pre-trained 2D flow matching model.

## B    RESULTS ON PANORAMIC GENERATION

We provide the panoramic generation results of OmniX from single-view image inputs, as shown in Figure 8. Our method is able to achieve high-quality and diverse image-to-panorama generation.

## C    RESULTS ON PANORAMIC COMPLETION

We provide the panoramic completion and guided panoramic perception results of OmniX from masked inputs and corresponding masks, as shown in Figure 9. Our method is able to achieve accurate and locally coherent completion and guided perception for panoramas.

## D    MORE ABLATION ANALYSIS AND DISCUSSION

**Impact of joint material modeling.**  VAEs for 2D latent flow matching models are trained on three-channel RGB inputs, and single-channel PBR material maps cannot be directly processed by the model. As a result, existing methods Kocsis et al. (2024; 2025) concatenate roughness and metallic together with an additional 0-channel for three-channel input. We explore the impact of different PBR material input arrangements on visual perception performance, as shown in Table 6. We find that that directly concatenating PBR material maps in the channel dimension is suboptimal, resulting in poor performance and blurred prediction results. A better practice is to jointly model PBR materials in a cross-attention manner.

Table 6: **Impact of joint PBR material modeling.** We utilize PanoX-Test and PanoX-OutDomain together as the evaluation set to comprehensively cover both in-domain and out-domain scenarios.

| Settings | Roughness | | Metallic | | Average | |
|---|---|---|---|---|---|---|
| | PSNR↑ | LPIPS↓ | PSNR↑ | LPIPS↓ | PSNR↑ | LPIPS↓ |
| joint (concat.) | 17.660 | 0.350 | 24.575 | 0.323 | 21.118 | 0.337 |
| joint (cross-attn.) | 17.427 | 0.340 | **25.425** | **0.138** | **21.426** | **0.239** |
| independent | **18.162** | **0.329** | 24.643 | 0.153 | 21.403 | 0.241 |

**Impact of joint geometry modeling.** Euclidean distance maps and normal maps are strongly correlated, so intuitively modeling them jointly should lead to improved performance. However, as shown in Table 7, such joint geometry modeling does not bring positive performance gains for the prediction of either modality. This may be because the model fails to learn the geometric relationship between distance and normal vectors from the limited training data.

Table 7: **Impact of joint geometry modeling.** We utilize PanoX-Test and PanoX-OutDomain together as the evaluation set to comprehensively cover both in-domain and out-domain scenarios.

| Settings | Distance | | | | Normal | | | |
|---|---|---|---|---|---|---|---|---|
| | AbsRel↓ | δ-1.25↑ | MAE↓ | RMSE↓ | Mean↓ | Median↓ | 5°↑ | 30°↑ |
| joint (cross-attn.) | 0.163 | 0.787 | 1.113 | 5.348 | 20.800 | 11.950 | 0.227 | 0.767 |
| independent | **0.155** | **0.808** | **1.084** | **5.347** | **19.917** | **10.992** | **0.249** | **0.779** |

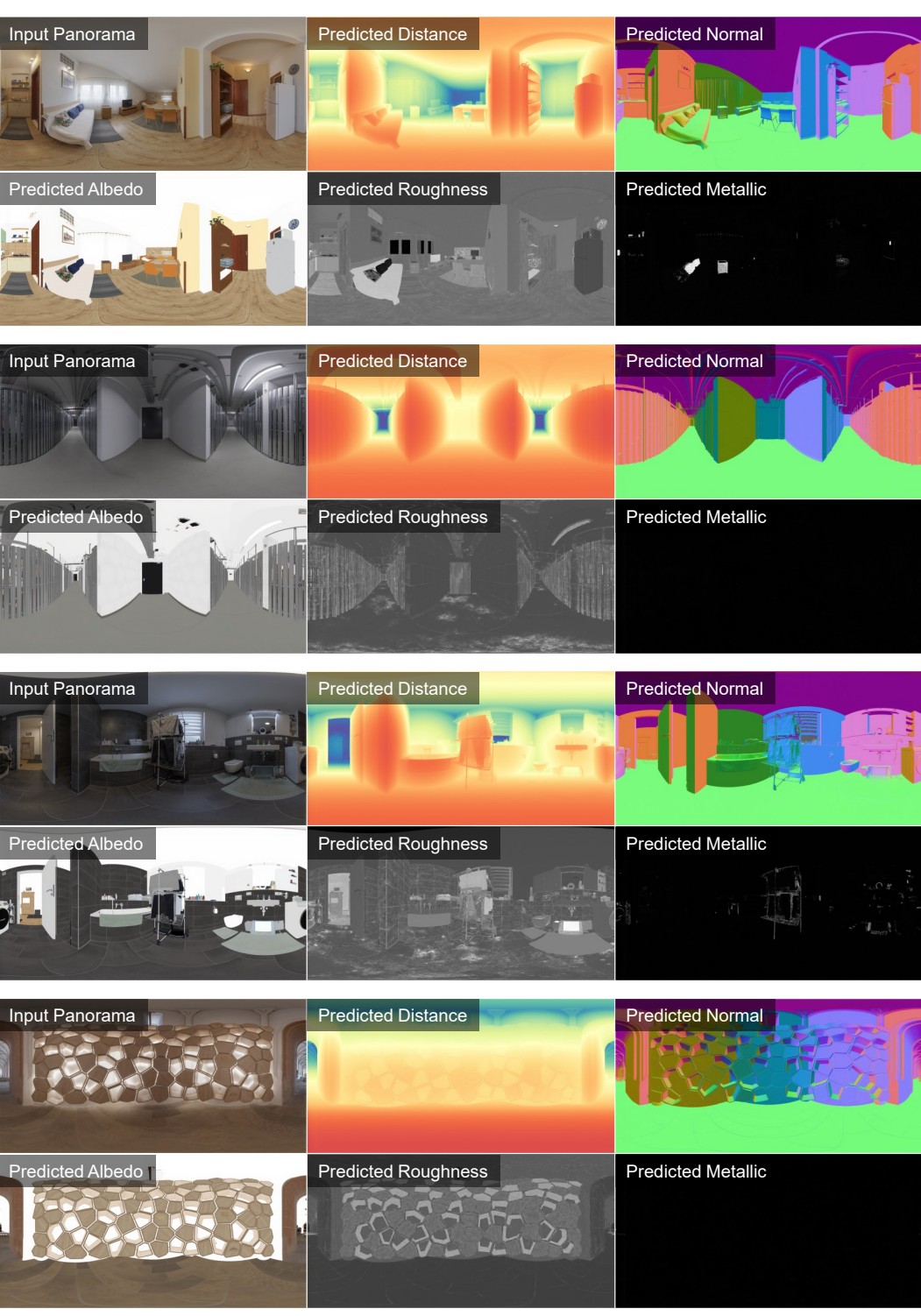

Figure 7: Panoramic perception results of OmniX on in-the-wild images. Our method demonstrates excellent generalization performance on unseen images.

864
865
866
867
868
869
870
871
872
873
874
875
876
877
878
879
880
881
882
883
884
885
886
887
888
889
890
891
892
893
894
895
896
897
898
899
900
901
902
903
904
905
906
907
908
909
910
911
912
913
914
915
916
917

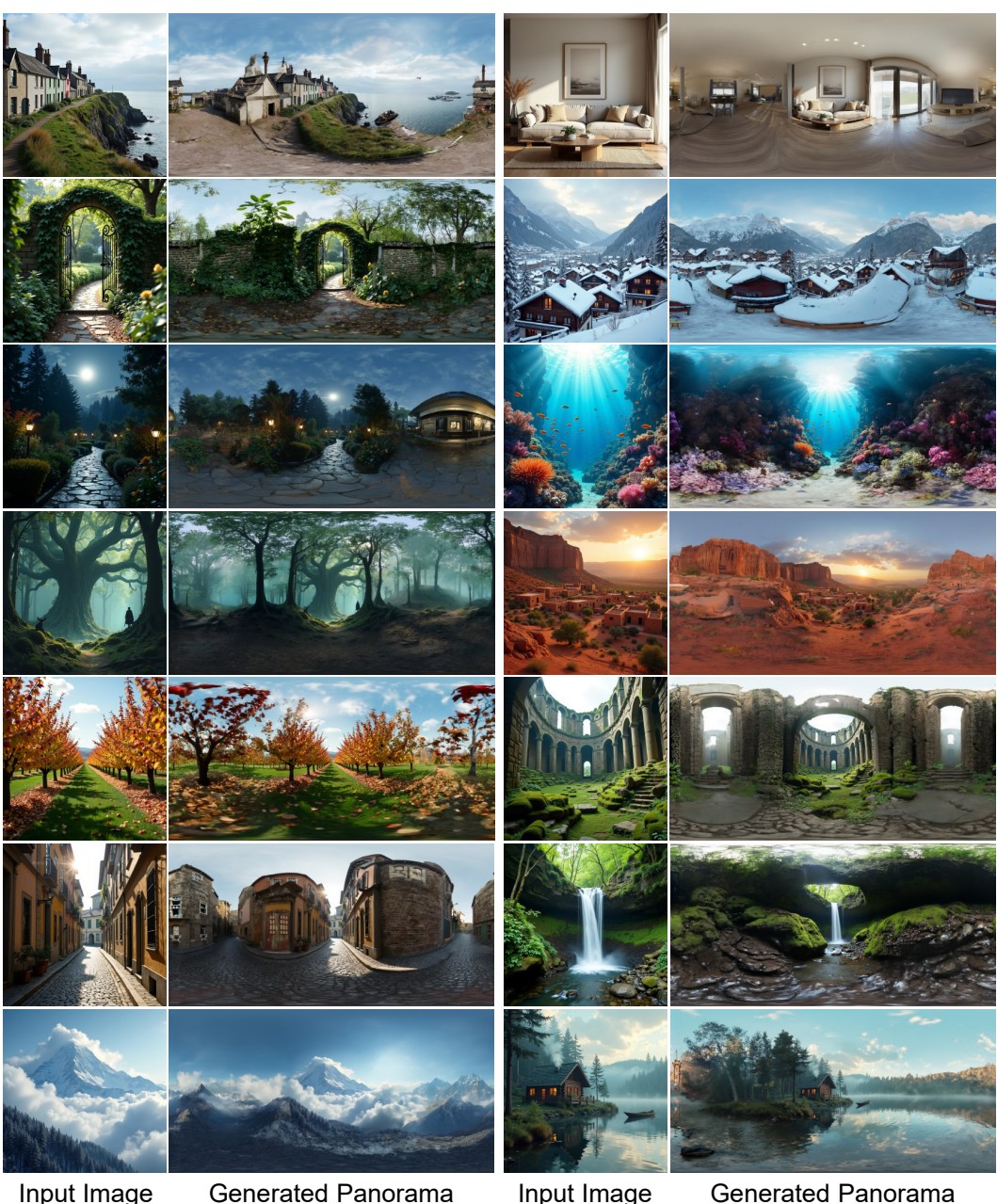

Input Image     Generated Panorama     Input Image     Generated Panorama

Figure 8: Panorama generation results of OmniX given a single image input. Note that the input single-view image is generated by Flux.1-dev (Labs, 2025).

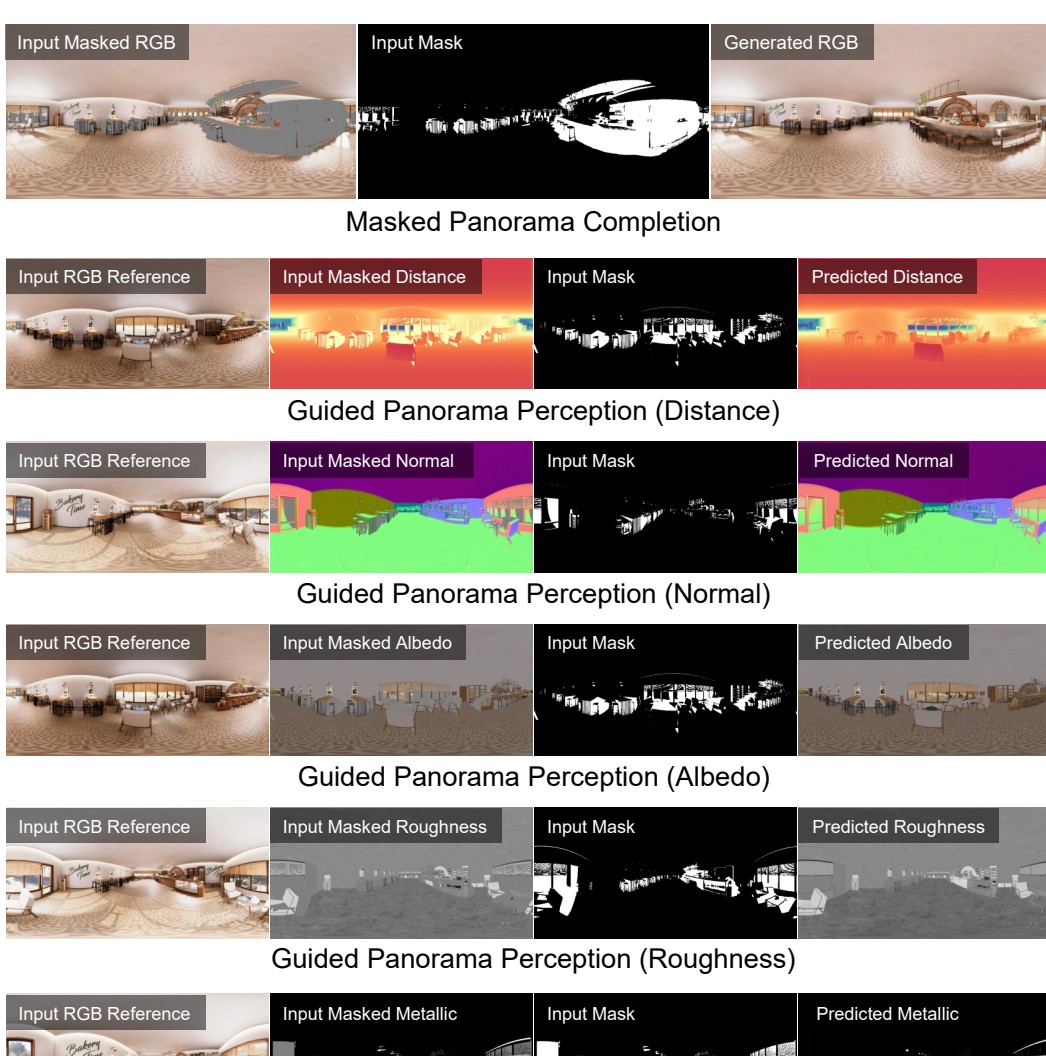

Figure 9: Panorama completion and guided panoramic perception results of OmniX. Given masked inputs and corresponding masks, OmniX is able to generate accurate and locally coherent results for masked areas. For guided panoramic perception, extra RGB references are input to ensure that the prediction results are consistent with RGB references.

# E    LIMITATIONS

Our method is built on top of pre-trained 2D flow matching models and thus inherits their shortcomings such as slow training and inference efficiency. In addition, OmniX's prediction of Euclidean distance is still not accurate enough, resulting in bumpy reconstructed 3D surfaces, which affects the subsequent PBR rendering effect. We also empirically observe that OmniX-Pano2Metallic, used for metallic prediction, performs poorly in generalization. This is partly due to the scarcity of panoramic PBR material data for training. Furthermore, the significant differences between neural rendering (*i.e.*, 2D generative modeling) and PBR rendering may indicate that pre-trained 2D image priors have limited benefits for PBR material estimation.

# F    USE OF LARGE LANGUAGE MODELS (LLMS)

We employ LLMs solely as a tool for language editing and improving the clarity of our writing. They did not contribute to research ideation, data analysis, or the development of methods. No significant role was played by LLMs in this work.

