# OpenReview forum: "OmniX: From Unified Panoramic Generation and Perception to Graphics-Ready 3D Scenes"
_ICLR.cc/2026/Conference — ICLR 2026 Conference Withdrawn Submission_

### Official Review · Reviewer_YcLu · 2025-10-27

**Soundness:** 3
**Presentation:** 3
**Contribution:** 3
**Rating:** 4
**Confidence:** 4

**Summary:**

The paper investigates graphics-ready multimodal generation for 3D scenes within a panoramic synthesis framework. The authors extend a DiT-based architecture with a dedicated conditioning branch and train separate conditional models to generate multiple physically meaningful modalities (e.g., normals, albedo) from an input RGB panorama. The approach is evaluated against strong baselines for RGB→X translation and demonstrates improved performance. Additional qualitative studies on in-the-wild panoramas and downstream geometry estimation further support the method’s versatility. Finally, the authors illustrate the practical utility of their generated modalities through physics-based simulation and relighting applications.

**Strengths:**

1. I like their motivation to infer multiple physically grounded modalities for enhanced scene generation and understanding. Extending this exploration to panoramic imagery is particularly relevant, as it can greatly benefit embodied perception and related downstream tasks.

2. The demonstrated downstream applications, such as relighting and physics-based simulation, are strong showcases of the practical value of the generated modalities. It is exciting that these capabilities are enabled directly by their multimodal DiT framework.

3. The proposed approach appears simple, flexible and extensible to many other modalities for physical simulation.

**Weaknesses:**

1. Insufficient analysis on performance gains: While the proposed model outperforms baseline methods, the paper does not clearly articulate why the design leads to better results. The architecture appears to be a relatively straightforward multimodal DiT extension, and it remains unclear which component contributes most to the improvements. For instance, is the gain primarily from leveraging flow-matching priors, conditioning design, or other architectural nuances? A deeper analysis or ablation would help justify the claimed advantages.

2. Practical limitations of the separate-adapter design: The separate-branch architecture raises concerns on its practicality in real world:
 (1) It significantly increases parameter count (effectively doubling the model)
 (2) It incurs higher memory cost during inference due to dual parallel branches
 (3) It introduces added engineering complexity, requiring distinct hyperparameters for multiple adapter modules (e.g., LoRA)
Without evidence of scalability benefits, it is uncertain whether this design is practical in real-world systems.

3. Lack of details on how baseline methods are prepared. For example, what dataset do their flow-matchin gmodel pre-trained on? How do they ensure a fair comparison across baseline methods?

4. The novelty of the proposed architecture

**Questions:**

1. It would be great to provide additional details on how the baselines methods are prepared. It is unclear why their method gives the strongest result in Table 2.

2. Could you clarify how's your conditional generation, i.e., separate-adaptor, compared with ControlNet related approach?

---

### Official Review · Reviewer_JjYH · 2025-10-30

**Soundness:** 3
**Presentation:** 3
**Contribution:** 2
**Rating:** 4
**Confidence:** 4

**Summary:**

OmniX presents a unified framework that leverages 2D generative priors for panoramic perception, generation, and completion, enabling the creation of graphics-ready 3D scenes suitable for PBR rendering and simulation. By introducing a cross-modal adapter and a large-scale multimodal panorama dataset, the framework achieves realistic and physically consistent 3D scene generation from 2D panoramas.

Overall, this work is the first to explore multimodal generation in the panoramic domain. However, the proposed methodology itself is not particularly novel, as similar techniques have been widely applied in perspective image generation. In addition, the paper lacks both quantitative and qualitative evaluations on other panoramic datasets (or in the wild datasets), which limits the generality and credibility of its claims.

Considering these factors, I would rate this paper as marginally below the acceptance threshold in my initial assessment.

**Strengths:**

1. This work makes a valuable contribution to panoramic perception by providing a synthetic dataset that supports unified panoramic perception training also a model to do multimodal generation in the panoramic domain.
2. The writing is clear, well-organized, and makes the paper easy to follow throughout.

**Weaknesses:**

Lack of Novelty:
The proposed methodology lacks sufficient novelty, as similar techniques have already been widely explored in the field of perspective image generation [1]. Essentially, the method appears to adapt existing approaches to the panoramic domain without introducing substantial methodological innovation.

Limited Comparison and Evaluation:
The proposed method is evaluated only on the authors’ self-constructed PANOX dataset. Although the dataset is divided into training, validation, and out-of-distribution (OOD) subsets, all samples are generated using the same pipeline. Due to the synthetic nature of the data and the use of a game engine, I assume the textures and visual characteristics across subsets remain similar. I recommend the authors evaluate their approach on publicly available datasets such as HM3D and Replica, and report results on panoramic-to-depth prediction tasks to demonstrate generalization.

Furthermore, the authors claim that their method enables graphics-ready 3D scene generation, but such results are not presented in the main paper. To substantiate this claim, I suggest including the final 3D reconstruction results for at least four example scenes in the paper.

[1] Ke B, Qu K, Wang T, et al. “Marigold: Affordable Adaptation of Diffusion-Based Image Generators for Image Analysis.” arXiv preprint arXiv:2505.09358, 2025.

**Questions:**

Is the proposed approach try to handle all tasks with one unified model instead of task-specific LoRA models?

---

### Official Review · Reviewer_P6Qy · 2025-11-01

**Soundness:** 3
**Presentation:** 3
**Contribution:** 2
**Rating:** 4
**Confidence:** 3

**Summary:**

This paper introduces OmniX , a unified framework that repurposes pre-trained 2D flow-matching models (e.g., FLUX.1-dev ) to address three core panoramic tasks: perception (RGB→X, where X includes depth, normal, albedo, roughness, and metallic), generation (image-to-panorama), and completion (masked panorama inpainting).  The framework leverages a cross-modal adapter structure (based on LoRAs) to reuse 2D generative priors without extensive modifications to the pre-trained weights. This enables the construction of "graphics-ready" 3D scenes compatible with physically-based rendering (PBR), relighting, and physics-based simulations.

**Strengths:**

1. Experiments on the PanoX, Structured3D, and HDR360-UHD datasets demonstrate that OmniX outperforms state-of-the-art (SOTA) methods in panoramic intrinsic decomposition. For instance, it achieves an albedo PSNR of 17.755 compared to 10.906 from DiffusionRenderer , while also achieving competitive performance in geometric estimation.


2.  The paper proposes a novel unified paradigm, OmniX , which effectively integrates panoramic perception, generation, and completion into a single 2D flow-matching-based framework. It reuses 2D priors without modifying pre-trained model weights. Compared to existing 2D lifting methods that are limited in scope (e.g., ImmerseGAN for appearance only, PhyIR for indoor geometry only), OmniX achieves an end-to-end connection from "appearance and intrinsic properties" to "3D scenes." This marks the first time that 2D-driven 3D generation can directly interface with PBR workflows, thereby simplifying 3D scene construction and enhancing its practical utility.

3. The authors constructed the PanoX panoramic dataset, the first of its kind to cover five indoor and three outdoor scene types with dense geometric and material annotations (as detailed in Table 1). Containing over 10,000 instances and 60,000 multimodal images generated via Unreal Engine 5, PanoX provides a much-needed benchmark for panoramic perception research and is poised to significantly benefit future work in the field.

**Weaknesses:**

1. The paper's geometric validation is superficial. It relies solely on 2D metrics (e.g., AbsRel, angular error) for depth and normal maps, which fails to prove the model has learned true 3D spatial information.

2. The work lacks crucial 3D-level validation, such as:
- Multi-view geometric consistency tests (e.g., verifying an object's size and shape remain consistent from different viewpoints).
- Analysis of learned geometric features to show the model understands distinct 3D shapes.

3. Additionally, the benefit of "introducing a camera ray" is not clearly justified, as the paper provides no quantitative data to verify its impact on 3D coordinate accuracy.

4. The connection between the model's geometric errors and the final 3D scene quality is unclear. The paper reports a depth error (AbsRel 0.158) but does not specify the resulting mesh defects (e.g., holes, vertex deviation).
It also fails to compare the final rendering quality of its meshes against those from other methods. This leaves the central claim—that these imperfect meshes are still suitable for PBR and physics workflows—unsupported and seemingly contradictory.

**Questions:**

The claim of "reusing 2D generative priors" is ambiguous. The manuscript does not explain how these priors are transferred across different tasks (e.g., whether priors from the generation task actually benefit the perception task). This claim could be substantially strengthened by including:

- A qualitative analysis (e.g., visualizing attention maps to show feature reuse).
- An ablation study comparing the performance of pre-trained weights versus random initialization.

---

### Official Review · Reviewer_7dYr · 2025-11-02

**Soundness:** 3
**Presentation:** 3
**Contribution:** 3
**Rating:** 6
**Confidence:** 4

**Summary:**

This paper proposes OmniX, a unified framework for panoramic generation, perception, and completion, built upon pre-trained 2D flow matching models. By introducing a cross-modal adapter structure, the framework flexibly supports multiple inputs (e.g., RGB, mask, camera rays) and outputs (e.g., depth, normal, PBR materials) without significantly altering the original model weights, effectively leveraging 2D generative priors to enhance panoramic task performance. To support model training, the authors constructed the PanoX dataset, which contains a large number of indoor and outdoor panoramic images with dense geometry and material annotations, filling the gap for high-quality data in this field. Experiments demonstrate that OmniX achieves state-of-the-art or competitive performance in intrinsic image decomposition (e.g., albedo, roughness, metallic) and geometry estimation (depth, normal) tasks, and showcases its effectiveness in image-to-panorama generation and mask-guided completion. Ultimately, by converting the prediction results into 3D meshes and assigning PBR materials, it enables the automatic construction of graphics-ready 3D scenes suitable for physically-based rendering, relighting, and simulation.

**Strengths:**

1.  This paper proposes a unified framework, OmniX, which repurposes pre-trained 2D flow matching models for panoramic image generation, perception, and completion tasks, demonstrating strong versatility and extensibility.
2. OmniX constructs a high-quality, multimodal panoramic dataset, PanoX, covering both indoor and outdoor scenes and providing dense geometry and material annotations, addressing a data gap in the current field.
3. This paper further proposes and compares various cross-modal adapter architectures. Experiments demonstrate that the Separate- Adapter structure significantly enhances multi-task performance while preserving the weight distribution of the pre-trained model.
4. Conducts thorough comparisons with state-of-the-art methods on multiple tasks (e.g., intrinsic image decomposition, geometry estimation), demonstrating superior performance, particularly excelling in material estimation.

**Weaknesses:**

1. Despite excellent performance in material estimation, the method still falls short of specialized approaches (e.g., MoGe) in depth estimation. The reconstructed 3D surfaces exhibit unevenness, which adversely affects subsequent rendering quality.
2. The authors note that the metallic prediction model has weak generalization capability. This is partly due to the scarcity of metallic material samples in the training data, also reflecting the limitations of 2D image priors for PBR material estimation.
3. The model struggles with maintaining seam continuity in ERP-format panoramic images, requiring post-processing techniques (e.g., horizontal blending) for mitigation. This indicates room for improvement in the model's understanding of panoramic structure.
4. Experiments on joint modeling of geometry (depth and normal) and materials (roughness and metallic) did not yield significant performance gains for either modality. This suggests that the intrinsic relationships between these multimodal signals have not been fully exploited.

**Questions:**

My primary consideration is the advantage of Omni generating multimodal data simultaneously compared to generating multimodal data separately, preferably supported by relevant experimental evidence.

---

### Note · Authors · 2025-11-12

**Comment:**

We sincerely appreciate the reviewers for their valuable feedback and constructive comments. While we have decided to withdraw our paper at this time, we are grateful for the time and effort they dedicated to evaluating our work. We will carefully consider their suggestions and make thorough revisions to improve the manuscript.

**Withdrawal Confirmation:**

I have read and agree with the venue's withdrawal policy on behalf of myself and my co-authors.